

# Separating the albedo reducing effect of different light absorbing particles on snow using deep learning

Lou-Anne Chevrollier[1], Adrien Wehrlé[2], Joseph M. Cook[1], Norbert Pirk[3], Liane G. Benning[4], Alexandre M. Anesio[1], and Martyn Tranter[1]

[1]Department of Environmental Science, iClimate, Aarhus University, Roskilde, Denmark
[2]Institute of Geography, University of Zürich, Zürich, Switzerland
[3]Department of Geosciences, University of Oslo, Oslo, Norway
[4]German Research Centre for Geosciences, GFZ, Potsdam and Germany Department of Earth Sciences, Free University of Berlin, Berlin, Germany

**Correspondence:** Lou-Anne Chevrollier (lou.chevrollier@envs.au.dk)

**Abstract.** Several different types of light absorbing particles (LAPS) darken snow surfaces, enhancing snow melt on glaciers and snowfields. LAPs are often present as a mixture of biotic and abiotic components at the snow surface, yet methods to separate their respective abundance and albedo-reducing effects are lacking. Here, we present a new optimisation method enabling the retrievals of dust, black carbon and red algal abundances as well as their respective darkening effects from spectral albedo. This method includes a deep learning emulator of a radiative transfer model (RTM), and an inversion algorithm. The emulator alone can be used as a fast and lightweight alternative to the full RTM with the possibility to add new features, such as new light absorbing particles. The inversion method was applied to 180 ground field spectra collected on snowfields in Southern Norway, with a mean absolute error on spectral albedo of 0.0056, and surface parameters that closely matched expectations from qualitative assessments of the surface. The emulator predictions of surface parameters were used to quantify the albedo reducing effect of algal blooms, mineral dusts and dark particles represented by black carbon. Among these 180 surfaces, the albedo reduction due to light absorbing particles was highly variable and reached up to 0.13, 0.21 and 0.25 for red algal blooms, mineral dusts and dark particles respectively. In addition, the effect of a single LAP was attenuated by the presence of other LAPs by up to 2-3 times. These results demonstrate the importance of considering the individual types of light absorbing particles and their concomitant interactions for forecasting snow albedo.

## 1 Introduction

Snow albedo is a key component of the Earth's radiation budget, and therefore it is important to both understand and model the drivers of snow albedo variability to project future snow melt and climate (Flanner et al., 2011). In particular, the darkening effect of light absorbing particles (LAPs) constitutes a major source of uncertainty for snow albedo in regional and global climate models (Skiles et al., 2018; Dumont and Tuzet, 2022). This uncertainty is notably due to the variability in the apparent optical properties of LAPs, which determine their impact on surface albedo. The apparent optical properties depend on microscopic and macroscopic parameters such as the size distribution, shape and refractive index of the particles, the mixing state



of the particles with the snow grains, the distribution and abundance of the particles in the snowpack, the shape and size of the snow grains, the surface roughness, the snowpack depth and illumination conditions (Flanner et al., 2021; He, 2022). It is clearly challenging to both measure all these parameters in the field and to constrain all of these parameters in models. Hence, uncertainties in field measurements and model parameterisations are propagated in forward modelling experiments aiming at quantify LAPs impact on snow surface albedo.

By contrast, inverse modelling approaches allow to study the impact of LAPs in snow directly from their measured apparent optical properties instead of prescribing all the above parameters, circumventing some of the uncertainty associated with forward modelling experiments. In addition, inverse methods offer a remote-only approach to detecting and quantifying LAPs in hard-to-access areas and/or over spatial scales that are too large to cover on foot. Inverse methods using physically based models have been developed to retrieve the abundance and impact of LAPs using look-up tables (Donahue et al., 2023; Chevrollier et al., 2023, e.g.) or simplified analytical formulas (Dumont et al., 2017; Kokhanovsky et al., 2021, e.g.), but these methods are limited in the number of parameters that can be accounted for. They therefore focus on a single LAP type, or merge all LAPs together. In particular, methods using look-up tables are advantageous in that they compare observations with model outputs directly, but the size of look-up tables grows exponentially as new parameters are included and hence they are usually not adapted to discriminate between LAPs. To our knowledge, only one study has developed an inverse method to discriminate between biotic and abiotic LAPs, based on Gaussian processes and optimal estimation (Bohn et al., 2021). Yet, consideration of LAPs separately is necessary because (1) each type of LAP impacts snow albedo differently, (2) the impact of a given LAP depends on the presence of other LAPs (Skiles and Painter, 2018; Kaspari et al., 2014) and (3) their future presence, abundance and impact on snow surfaces depend on different processes, although feedbacks between LAPs can exist (McCutcheon et al., 2021; Harrold et al., 2018). For example, the presence of black carbon depends mainly on the burning of fossil fuels and forests (Bond et al., 2013), while the presence of dust and algae are more dependent on the climate, such as droughts, dust storms or environmental conditions sustaining microbial life (Skiles et al., 2018). Developing inverse methodologies discriminating between LAPs is therefore necessary to deepen our understanding of snow surfaces and the role of LAPs in future snowmelt, as well as improving and validating existing albedo models, in particular in their representation of snow algae (Flanner et al., 2021).

Here, we aim at building an efficient, robust and unbiased inversion scheme able to simultaneously retrieve red algal, mineral dust and black carbon abundances and darkening effects from spectral surface albedo. We first integrated new empirical optical properties for red snow algae as well as snow liquid water content (LWC) in the radiative transfer model (RTM) biosnicar, then built a a deep-learning emulator of the model and an inversion algorithm utilising this emulator. We present the performance and benefits of the emulator in comparison to the original RTM, explore the robustness of the inversion algorithm, and then apply this new method to field spectra collected in Norway. Finally, we discuss the potential of the method to be upscaled to airborne imagery.



## 2 Methods

### 2.1 Inversion scheme

Simulations of the open source RTM biosnicar (Cook et al., 2020), a python translation of the SNICAR model (Flanner et al., 2007), were generated (section 2.1.1) and used to build a deep learning model emulating the RTM (section 2.1.2). The emulator was then coupled to an optimising algorithm (2.1.3) to invert spectral albedo for surface properties, including the darkening effect of light absorbing particles. All analyses relied importantly on the python libraries numpy (Harris et al., 2020), pandas (pandas development team, 2024; Wes McKinney, 2010), tensorflow (Developers, 2024), keras (Chollet et al., 2015), matplotlib (Hunter, 2007), scipy (Virtanen et al., 2020), random, joblib, time, multiprocessing and scikit-learn (Pedregosa et al., 2011).

### 2.1.1 Input data

The RTM was parametrised with two snow layers of 0.02 and 100 m with varying physical and biological properties (LAPs concentrations, snow grain size, liquid water content, and solar zenith angle), representing snow as a granular medium with spherical grains. The upper layer contained black carbon, mineral dust and red snow algae, and the two snow layers had the snow same grain size and liquid water content. A 2 cm depth was chosen for the upper layer as this depth was used to quantify algal cells in recent field studies (Engstrom et al., 2022; Healy and Khan, 2023). The second layer was chosen as semi-infinite to remove the dependency of the spectral albedo on the underlying surface, hence the simulations do not represent very thin snowpacks. The illumination was set as direct beam mid-latitude summer irradiance for different solar zenith angles (SZA). The density was kept constant at 600 kg m$^{-3}$, which is justified because the snow grain size is here an effective optical grain size (Warren, 1982; Gardner and Sharp, 2010) that can cover realistic ranges of snow specific surface area for melting snow (1-10 m$^2$ kg$^{-1}$; (Dumont et al., 2017; Tuzet et al., 2020)). The parameter ranges for the simulations are given in Table 1. These ranges were not linear but rather manually designed by creating homogeneous meshes, accounting for radiative transfer non-linearity.

| Red snow algae (cells mL$^{-1}$) | Dust (ppb) | Black carbon (ppb) | Liquid water content (%) | Grain size (μm) | SZA |
|---|---|---|---|---|---|
| 0-1.5 × 10$^5$ | 0-2 × 10$^6$ | 0-2 × 10$^3$ | 0-15 | 350-3000 | 35-60 |

**Table 1.** Parameter ranges used in the RTM simulations for the training of the emulator

The liquid water content (LWC) in snow was implemented in the RTM following the methodology of Flanner et al. (2021) and Donahue et al. (2022). First, the optical properties of 5000 individual water and ice spheres with log-spaced radii ranging from 0.05 to 10000 μm were generated from water and ice refractive indices using the miepython package (Prahl, 2023). The ice refractive index was the one compiled from Picard et al. (2016) and Warren and Brandt (2008) as described in Flanner et al. (2021), and the water refractive index was a compilation between the dataset at 0°C from Rowe et al. (2020) beyond 0.7 μm, and the dataset from Segelstein (1981) below 0.7 μm. Then, the optical properties of lognormally distributed water and ice spheres were computed for a given effective radius (Flanner et al., 2021), and mixed using a volume weighed average depending





on the LWC (Donahue et al., 2022). For the light absorbing particles, the optical properties of black carbon (BC; uncoated) and dust (10-50 μm) were directly available in the RTM from previous work (Flanner et al., 2012; Skiles et al., 2017; Flanner et al., 2021). For red snow algae, new empirical optical properties were integrated in the RTM. The single scattering properties

of red snow algae were directly derived empirically instead of using Mie theory to avoid making assumptions on pigmentation, refractive index and size distribution of the algae. The absorption coefficient was measured using the filter-pad transmission method (Stramski et al., 2015), the extinction coefficient was measured on a regular spectrophotometer from normal-normal transmittance (Kandilian et al., 2016), and the asymmetry parameter was assumed constant at 0.96 (Dauchet et al., 2015). The absorption coefficient was corrected from the instrument scattering biases with a baseline at 0.8 μm, a smoothing in the

0.3-0.5 μm range where the signal was noisy, and curve sharpening where scattering effects distorted the spectrum. The optical properties are representative of algal bloom from the Greenland ice sheet where the samples were collected (Chevrollier et al., 2023), and are assumed to generalise to all red algal blooms.

The RTM outputs albedo spectra in the 0.205-4.995 μm range with steps of 0.01 μm, and only the spectral range of 0.295-2.405 μm was used here because (1) the sun irradiance is negligible below 0.295 μm, (2) hyperspectral/multispectral sensors rarely

detect signals above 2.5 μm, (3) biosnicar produces discontinuities around 2.5 μm, and (4) reducing the spectral range allows to reduce the training time of the emulator (see 2.1.2). In addition, the simulated albedos were converted to float16 to reduce the size of the look-up table files. The RTM-derived dataset contained 5,827,464 simulations, with 6 input features (grain size, LWC, BC concentration, dust concentration, red algal concentration, SZA) and a 212 bands albedo target. 90% of the dataset was used for training and 10% for testing the emulator. Prior to training, the input features were normalised to values between

0 and 1 to avoid biases in emulator training that would focus the learning on the parameters with the largest values.

### 2.1.2   Neural network

A feed forward neural network was designed using the Keras library (Chollet et al., 2015). The number of layers and neurons per layer, which determines the number of weights, was optimised from repeated runs of the Keras tuner bayesian optimisation, which searches for the model architecture with the lowest associated error among prescribed architectures. The chosen network

had 12 hidden layers with different numbers of neurons for a total of 266,031 trainable parameters. The activation function for all layers was the exponential linear unit (elu), chosen to avoid a vanishing gradient problem, except for the last one, for which the activation function was linear. The performance of the model was monitored with three evaluation metrics: the mean absolute error (MAE), the mean absolute percentage error (MAPE), and the mean squared error (MSE). The loss of the model training was the mean absolute error, as it yielded the lowest values for all built-in evaluation metrics of Keras.

The optimiser of the neural network was the Adaptive Moment estimation (Adam) optimiser with an adaptive learning rate decreasing exponentially, starting from 0.001. The model was trained for 200 epochs with a batch size of 256. For each epoch, the training dataset was shuffled and 10% was used for validation.





### 2.1.3 Inversion algorithm

The inversion algorithm was a gradient-descent algorithm built using the GradientTape tensorflow object, following the methodology of Jouvet et al. (2021). The goal of the algorithm is to iteratively minimise the cost between a target spectrum and the neural network's predicted spectrum in order to find the closest albedo curve and associated surface properties. The gradient of the cost is computed (and stored) with regards to each variable for 1000 iterations, except for the SZA which was fixed because the sun angle can be calculated prior inversion for a given spectrum. The gradient is then applied to the variables using the built-in Adagrad optimiser from Keras, and the values of the features are kept positive to prevent the gradient-descent from entering non-physical parameter spaces, i.e. negative surface properties. To avoid the retrieval of local minima, a pseudo-random sampling approach was implemented where the inversion was computed for 20 different initialisation points sampled randomly, in order to increase the chances of reaching a cost as low as possible by exploring different areas of the parameter space. The algorithm then selects the retrieval with the lowest cost out of the 20 initialisations as the final retrieval.

### 2.2 Ground spectroscopy

Hemispherical-Conical Reflectance Factor (HCRF) measurements were collected on snowfields around the Finse Alpine Research Center in summer 2023. In total, 185 measurements were collected using an ASD Fieldspec4 (spectral range 0.35-2.5 μm) and a black tripod, following the methodology of Cook et al. (2017). Each measurement was acquired with the bare fiber (field of view of 25 degrees) in 10 replicates, immediately (< 10s) after a reference spectrum was measured using a calibrated Spectralon panel. All measurements were performed at nadir view (viewing zenith angle $\theta = 0$ °) with the tripod oriented towards the sun to avoid shadow effects. Most (80%) of the measurements were taken for a solar zenith angle between 38 ° and 50 °, and the remaining (20%) were taken for a solar zenith angle below 59 °. The spectra were corrected for the panel spectral response as well as the step at 1 μm caused by the misalignment of the SWIR and NIR sensors in the ASD FieldSpec, following Painter (2011). For all spectra the step at 1 μm was however not visible, probably because the instrument was warmed up for at least an hour prior the measurements and the selected surfaces were homogeneous enough. 5 spectra were removed from the analysis because clear calibration errors were detected.

HCRF measurements have several benefits over using a cosine collector at our specific site: a) The measurements are faster to execute, making it easier to avoid error due to changing illumination conditions, b) the incoming irradiance is fully re-diffused by the reflectance panel, avoiding errors arising from the cosine property and levelling of the collector (Aoki et al., 2000), c) the restricted field-of-view allowed to select relatively homogeneous and flat surfaces, reducing the error arising from micro-topography and shadow effects. Since the measurements are theoretically not equivalent to spectral albedo, the impact of snow anisotropy on the inversion retrievals was estimated by applying the anisotropy reflectance factor (ARF) measured by Dumont et al. (2010) to the HCRF spectra. We selected the ARFs measured by Dumont et al. (2010) over a surface with high visible impurities content at nadir viewing to best fit the type of surface and measurement set up of this study, with incident radiation angles bounding the minimum and maximum solar zenith angle at the time of our measurements.





## 2.3 Albedo reduction and instantaneous radiative forcing calculations

Broadband albedo (BBA) was calculated as:

$$BBA = \frac{\sum \alpha(\lambda) \times I(\lambda)}{\sum I(\lambda)} \tag{1}$$

with $\lambda$ the wavelength in the spectral range of the emulator (0.295-2.405 μm), $\alpha(\lambda)$ the spectral albedo outputted by the emulator and I($\lambda$) the spectral irradiance from the RTM for a given SZA. The SZA corresponding to each HCRF measurement

was calculated using the pysolar python package (Stafford et al., 2023). The SZA was set at 50 ° for measurements taken under overcast conditions (48%), since it approximates the spectral albedo under diffuse irradiance (Wiscombe and Warren, 1980). The BBA reduction associated with a given LAP was calculated by differencing the BBA with and without the given LAP. The daily radiative forcing (W m$^{-2}$) was calculated by multiplying the BBA reduction with the 24h daily averaged shortwave incoming radiation, as measured with a four-component radiometer (CNR4, Kipp and Zonen, The Netherlands) at the local

weather station (Pirk et al., 2023).

## 3 Results and discussion

### 3.1 Performance, benefits and uncertainties of the inversion method

#### 3.1.1 Forward emulator

The neural network reproduced the original RTM simulations very accurately, with a mean absolute error (MAE), mean ab-

solute percentage error (MAPE) and mean squared error (MSE) on the spectral albedo of respectively $5.3 \times 10^{-5}$, 0.047 %, and $8.5 \times 10^{-9}$ both for the training and testing sets, indicating that the emulator does not overfit. The maximum MAE of the 5,827,464 samples of the training and testing sets was $5.8 \times 10^{-4}$, with 99% of the dataset predicted with a MAE below $8 \times 10^{-5}$ (Fig. 1a). The spectral residuals of the prediction associated with the highest MAE are below $5.0 \times 10^{-3}$ (Fig. 1b), and below $2.0 \times 10^{-5}$ for the lowest MAE (Fig. 1c).

The emulator has several benefits over and above running the full RTM: 1) it executes ∼30x faster; 2) it is a portable and lightweight Keras model occupying just 1MB of disk space, whereas the full RTM package occupies ∼1 GB of disk space due to the large optical property database and look up tables that come bundled with it; 3) it enables the exploration of quasi-continuous spaces for all parameters, in comparison to the original RTM python package which relies on the optical property database for example for the snow grain size or SZA, and 4) it is a Keras model that can be loaded into scripts written in several

popular programming languages (including Python, C, and FORTRAN), potentially enabling straightforward RTM integration into larger climate models. The emulator is therefore is a practical and efficient alternative to the full RTM for the majority of use cases.

The computation efficiency of the emulator in comparison to the original RTM grows when predicting several spectral albedos at once, but this improvement depends on the number of simulations and the use of multiprocessing. For example, the look-up

table of +5,000,000 spectral albedos produced in this study can be generated with fully vectorised operations within the neural





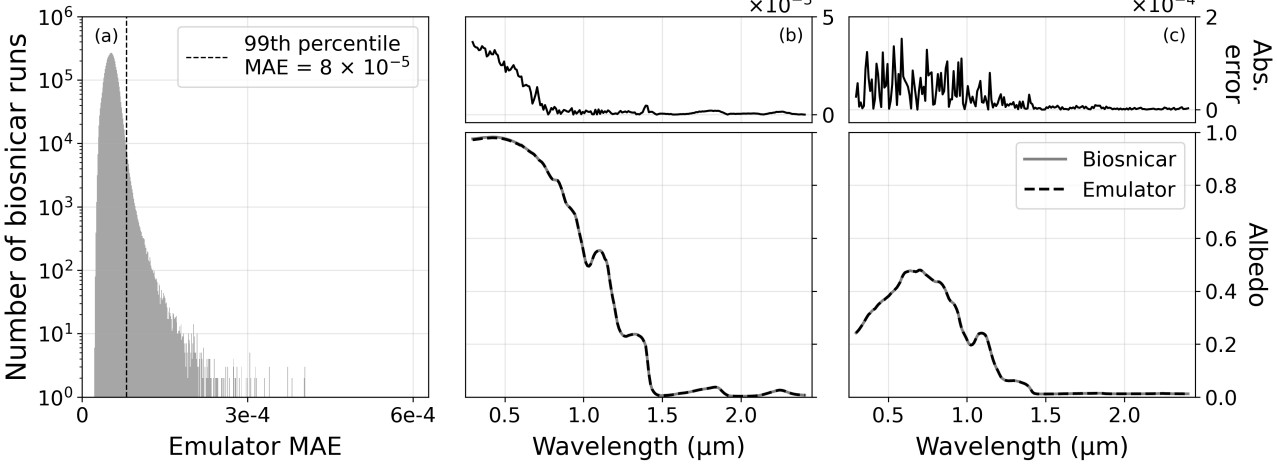

**Figure 1.** Performance of the emulator in reproducing the RTM simulations. (a): distribution of the mean absolute error (MAE) of the residuals between the original RTM model runs and the predictions of the emulator. Emulator-predicted vs. RTM-simulated albedos corresponding to the (b) the highest and (c) lower mean absolute residuals, with the associated spectral residuals above. Note that the scale on the MAE distribution is logarithmic.

network (ie, happen simultaneously in-memory), which makes the computation speed 4 orders of magnitude faster than running the original RTM on a single thread. In the context of this study, the computation efficiency of the emulator also offers the possibility to carry out spectral inversions in a reasonable time.

### 3.1.2 Inversion algorithm

The neural network was incorporated into an inversion algorithm that retrieves the surface parameters required by the RTM to best reproduce a spectrum given as input. The inversion algorithm was evaluated on an ideal dataset with known surface properties to assess its ability to converge to an optimal solution, when it exists. 2000 sets of surface properties were sampled pseudo-randomly and fed into the emulator to produce a synthetic dataset of 2000 spectral albedos, which were then inverted. The inversion algorithm reproduced the 2000 albedo curves with a mean absolute error (MAE) of $5 \times 10^{-7}$ and the retrieved

surface properties matched those of the synthetic dataset almost perfectly (Table 2), indicating that the algorithm is capable to reach an optimal solution if one exists. The standard deviation between the 20 different initialisation points for the albedo MAE, the algal concentration, the dust concentration, the BC concentration, the LWC and grain radius were respectively $1 \times 10^{-4}$, 497 cells mL$^{-1}$, $1.3 \times 10^4$ ppb, 14 ppb, 0.03 % and 4 μm. These variations are low and produce nearly undetectable effects on the spectral albedo in our model configuration, indicating that the algorithm is stable and reaches a similar solution

despite starting from a different and random point. At present, the inversion method is fast enough to be scaled to a satellite tile on a 16-cores machine (CPU: AMD Ryzen 7 7700X 8-Core Processor and RAM: trident Z5 neo (2*32G)). The method is





| Surface property | Algal concentration | Dust concentration | BC concentration | LWC | Grain radius |
|---|---|---|---|---|---|
| Mean Absolute Error (MAE) | 5 cells mL$^{-1}$ | 116 ppb | 0.1 ppb | $4 \times 10^{-5}$ % | 0.003 μm |
| Slope | 0.99996 | 0.999897 | 0.9998 | 0.9999991 | 0.999997 |
| Bias | 1.02 cells mL$^{-1}$ | 116.5 ppb | 0.09 ppb | $-1.52 \times 10^{-7}$ % | 0.005 μm |

**Table 2.** Mean error between the original and retrieved surface properties; slope and bias of the correlations between the original and retrieved surface properties.

hence efficient enough in speed and accuracy to open the possibility of separating the abundance and impact of LAPs at large scale.

## 3.2    Application to ground spectra

The inversion algorithm successfully reproduced ground HCRF spectra collected on snowfields in Southern Norway (N=180, average MAE = 0.0056, max. MAE = 0.0089), including the apparent optical properties of LAPs (Fig. 2). Figure 2 shows the retrievals for four different surface types with distinct signatures: (a) relatively clean old snow with low concentrations of LAPs, (b) red snow surface with high algal concentration, (c) brown snow surface with high dust concentrations and (d) dark snow surface with high BC concentrations. The model accurately reproduces the red algal pigment features, and the retrieved

concentrations of algae for the entire dataset covered the range of typically measured values (up to $1.2 \times 10^5$ cells mL$^{-1}$), with higher concentrations being associated with visibly redder surfaces (Fig. A1). The typical UV-absorbing feature of dust and the UV-VIS flattening signature of black particles were equally well reproduced (Fig. 2c, d). No measurements were performed to know whether BC was present in the snow and the retrieved BC concentrations are likely to represent dark particles of different origin, most of them coming from the vegetation around the snowfields. Therefore, BC is hereafter referred to as dark particles.

The standard deviation on the dust, algal and dark particle concentrations between the 20 random initialisations of the inversion algorithm was 514 ppb, 25 cells mL$^{-1}$ and 0.6 ppb respectively. This variability produces negligible effects on the spectral albedo, which shows that the uncertainties of the inversion method are primarily related to the assumptions on the field data rather than the randomness of the inversion algorithm. In particular, HCRF measurements are not strictly equivalent to hemispherical spectral albedo modelled by the RTM due to reflectance anisotropy (Cook et al., 2017), but no anisotropy

correction exists for the type of wet, rough, aged and heterogeneous snow surface targeted by this study. The anisotropy of snow is more pronounced beyond 1.4 μm (Aoki et al., 2000) and its effect is lower for measurements taken at nadir view and incident angles below 60 ° (Dumont et al., 2010). Sensitivity analyses indicate that the effect of anisotropy on the retrievals was different for each type of LAP and that this effect can be positive or negative depending on the LAP concentration and the incident angle (Fig. A2). The algal concentrations were the least affected by the anisotropy correction (Fig. A2). Beyond

anisotropy, the slope of the surface can have a significant effect on snow albedo and corrections exist if the spectral diffuse and direction partitioning of the incoming irradiance as well as the slope inclination and aspect have been measured (Picard et al., 2020). Future work could therefore include measurements of the slope and roughness of the surfaces, along with the LAPs





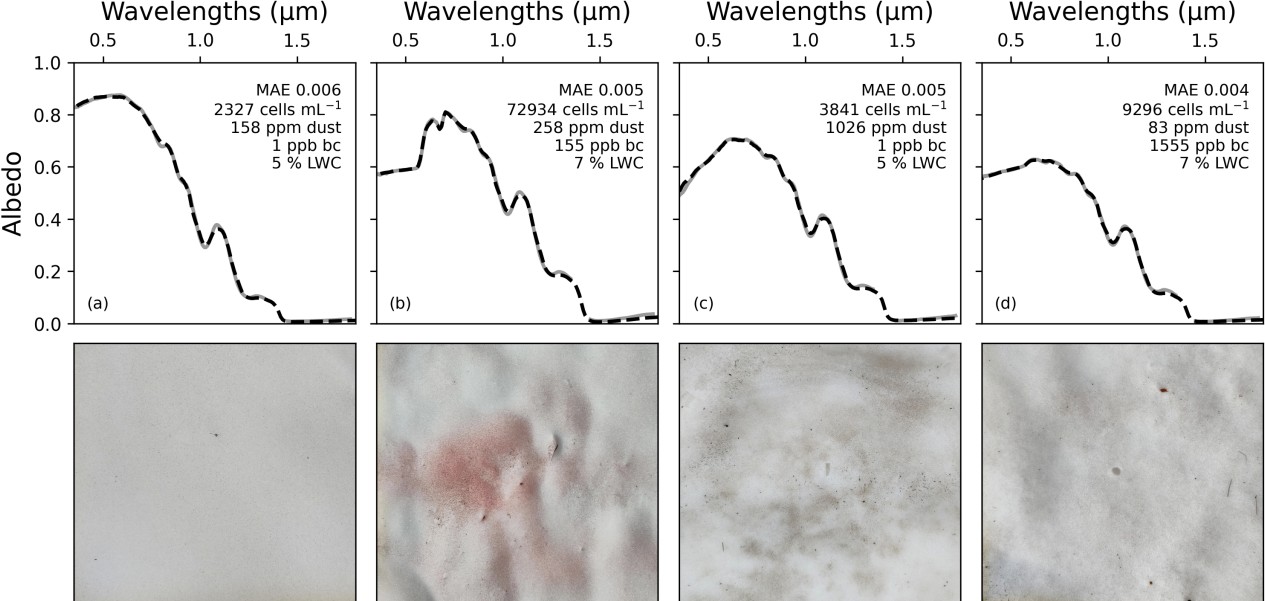

**Figure 2.** Field reflectances (gray) vs retrieved spectral albedos (dashed black) and associated surface properties for four different surface snow types: (a) relatively clean snow, (b) very red snow, (c) dust-loaded snow, and (d) dark particles-loaded snow. The scale of the images at the bottom is approximately 45x45 cm and are centered on the middle point of the footprint of the reflectance measurements.

concentrations, to distinguish between the error related to the LAPs abundance and the geometry of the surface. In this study, given the variability in the effect of anisotropy, the effort to select flat and homogeneous surfaces during sampling and keep

the solar zenith angle below 60 °, the nadir viewing, and the fact that our sample surfaces had varying physical configurations that would require specific ARFs, HCRF measurements are considered to be equivalent to hemispherical albedo for BBA and radiative forcing calculations presented herein.

The BBA reduction of LAPs ranged from 0.04 to 0.25 for the 180 data points, equivalent to a daily radiative forcing (RF) of 7-44 W m$^{-2}$ for the average daily summer illumination conditions at the field site (Fig. 3a), and 15-83 W m$^{-2}$ for the

sunniest day of the season. These values cover the range of reported values for the maximum daily radiative forcing of LAPs in Europe and North America (Skiles et al., 2018), and compare well with the recent estimation of 58 W m$^{-2}$ reported in Col du Lautaret, France (Tuzet et al., 2020). Red algal blooms alone reduced the albedo by up to 0.13 (mean: 0.05), dust reduced the albedo by up to 0.21 (mean: 0.02), and dark particles reduced the albedo by up to 0.25 (mean: 0.03). The albedo-reducing effect of each LAP was lower than if it was present alone, because the darkening caused by the presence of a LAP is lower

when the background surface is also darkened by other LAPs. In this case, the impact of algal blooms, dust and dark particles were on average 39, 62 and 45% lower, and up to 2.8, 3.2 and 2.6 times lower, than if each type of LAP had been present alone. These results align with previous observations for dust and black carbon (Skiles and Painter, 2018; Kaspari et al., 2014) and emphasise the need to study LAPs concomitantly.



The daily average RF caused by dust, red algae and dark particles were respectively up to 36 W m$^{-2}$ (mean: 4.0 W m$^{-2}$), 22 W m$^{-2}$ (mean: 9.0 W m$^{-2}$) and 44 W m$^{-2}$ (mean: 6.1 W m$^{-2}$) and reached up to 72 W m$^{-2}$, 44 W m$^{-2}$ and 87 W m$^{-2}$ during the sunniest day. In comparison, the dust-specific daily RF estimates from surfaces in the Upper Colorado River Basin were of the same order of magnitude, although higher as the surfaces were visibly more dust-loaded there (Skiles and Painter, 2018), and the red algal-specific daily RF was comparable to the effect of red blooms in Antarctica (13 W m$^{-2}$; Khan et al. (2021)). The maximum instantaneous radiative forcing (IRF) due to LAPs for the entire dataset was 337 W m$^{-2}$. The maximum dust IRF was 277 W m$^{-2}$, which is of the same order of magnitude to those reported in the Alps (up to 154 W m$^{-2}$; Di Mauro et al. (2015)) and in the Upper Colorado River Basin (525 W m$^{-2}$; Skiles and Painter (2018)). In comparison, the maximum algal IRF was 169 W m$^{-2}$, which closely matched the values reported for Alaska (up to 175 W m$^{-2}$; Ganey et al. (2017)), and the North Cascades at the end of July (156.9 W m$^{-2}$; Healy and Khan (2023)), but was lower than those reported in British Columbia at the end of July (up to 295 W m$^{-2}$; Engstrom et al. (2022)) and in the North Cascades at the beginning of July (359.95 W m$^{-2}$; Healy and Khan (2023)). Differences in maximum IRF for a given LAP can be due to a different LAP abundance, the presence of other LAPs at the surface, a different surface distribution, and/or a higher maximum irradiance. Finally, the 180 data points presented here are representative of small footprints of $\sim 0.1$ m$^2$ and were not specifically collected to be representative of larger surface areas, yet they serve as an illustration of the range of impact and relative contribution that LAPs can have. Indeed, the new method employed here enables the retrieval of surface properties that are otherwise not accessible, such as the separate impact of each type of light absorbing particle. This dataset therefore demonstrates the high variability of LAPs impact, emphasising the need to study them both separately and concomitantly, and highlighting the important role of biotic LAPs in snow surface albedo reduction (Fig. 3b). Eventually, other LAPs could be integrated to the model for it to be applicable to other locations where ashes, brown carbon, green algae or different types of dust are present on the snow surfaces.

## 4 Conclusions

The light absorbing particles darkening snow surfaces have varying apparent optical properties, challenging the quantification of their albedo reducing effect using forward models. Here we present an efficient, robust and portable method aimed at simultaneously retrieving the abundance of LAPs and their albedo-reducing effect on snow surfaces from spectral albedo measurements. It consists of a neural network emulating a radiative transfer model and an inversion algorithm. We found that the neural network added negligible error compared to using the full RTM and ran significantly faster. We used the neural network in inverse mode to retrieve the surface properties that best explain observed field spectra and calculate how much albedo change and radiative forcing can be attributed to red algae, dust or dark particles. The results highlight a wide range, magnitude, and co-dependency of LAP impacts, illustrating the subsequent need to study LAPs both separately and concomitantly. The high and stable performance of this inversion algorithm on field spectra indicates that it could also be applied to imagery obtained from drones or satellites to improve the diversity of information captured from regional scale mapping of snow.



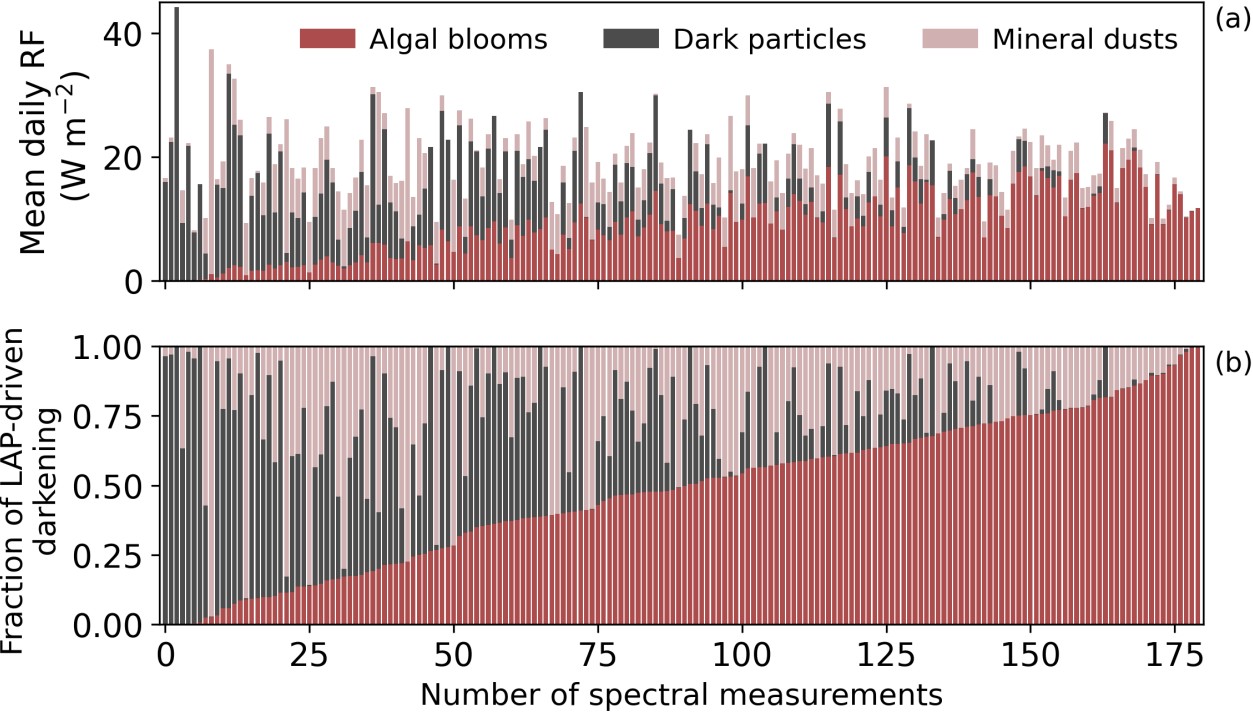

**Figure 3.** Radiative forcing (a) and contribution to LAP-driven albedo reduction (b) associated with the presence of each LAP for the 180 spectra analysed.

*Code and data availability.* The preprint is submitted with an archive including all the data and code required to reproduce the results and figures of this study: 1) the spectral data, 2) the AWS data, 3) the emulator and inversion code. The biosnicar version used is available at https://github.com/jmcook1186/biosnicar-py. Upon publication, the will be available through a zenodo archive and an online github repository.

*Author contributions.* Conceptualisation: LC, AW; Data collection: LC, AW, NP; Data curation: LC, AW, NP; Algorithm development: LC, AW; Biosnicar maintenance: LC, JC; Visualisation: LC, AW; Writing - original draft preparation: LC; Funding acquisition: LGB, AMA, MT; Writing - review and editing: All authors.

*Competing interests.* The authors declare that they have no conflict of interest.

*Acknowledgements.* This work is part of the project DeepPurple that has received funding from the European Research Council (ERC)
under the European Union's Horizon 2020 research and innovation programme (Grant agreement No. 856416). LC thanks Marie Dumont





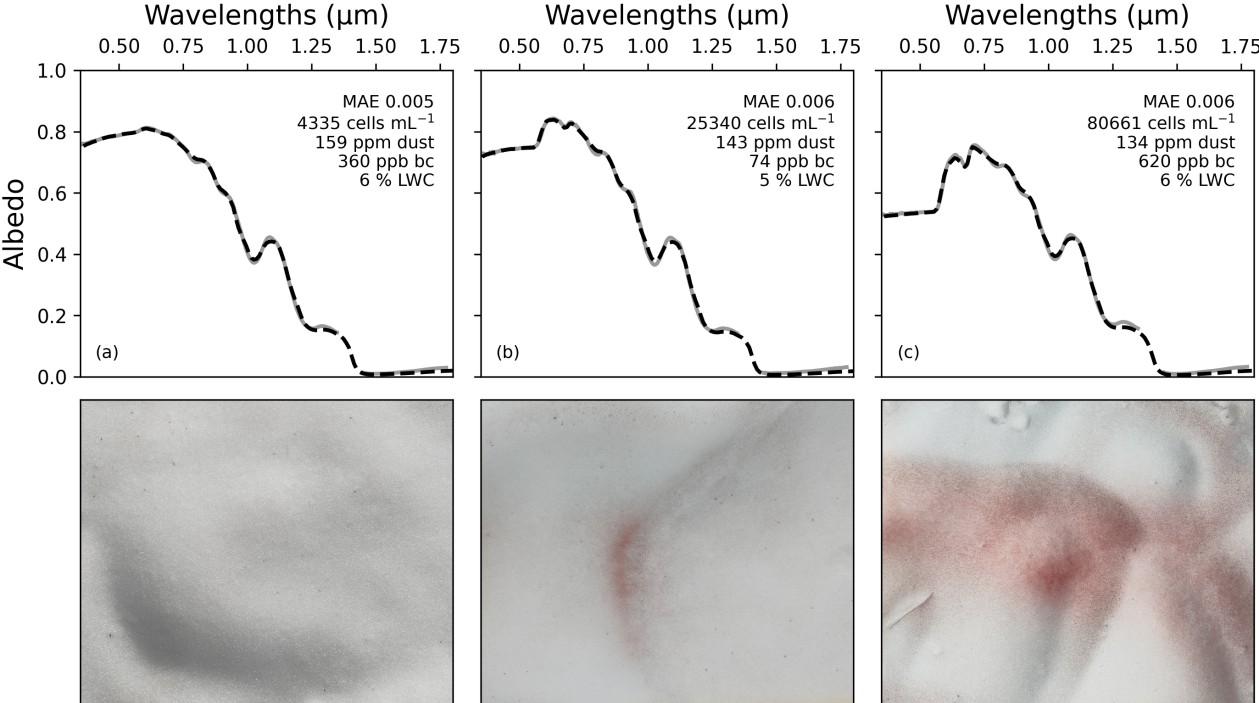

**Figure A1.** Field reflectances (gray) vs retrieved spectral albedos (dashed black) and associated surface properties for three surfaces visibly more and more red. The scale of the images at the bottom is approximately 45x45 cm and are centered on the middle point of the footprint of the reflectance measurements.

for guidance and suggestions related to the treatment of snow anistropy, Mark Flanner and Chloe A. Whicker-Clarke for the guidance to generate optical property files. LC and AW thank Guillaume Jouvet for guidance on the development of the inversion algorithm, Jens Ådne Rekkedal Haga for facilitating fieldwork logistics, the organisers of the Machine Learning in Glaciology Workshop (https://github.com/ Machine-Learning-in-Glaciology-Workshop) and the Finse Alpine Research Center, which is maintained by the University of Oslo and the University of Bergen.




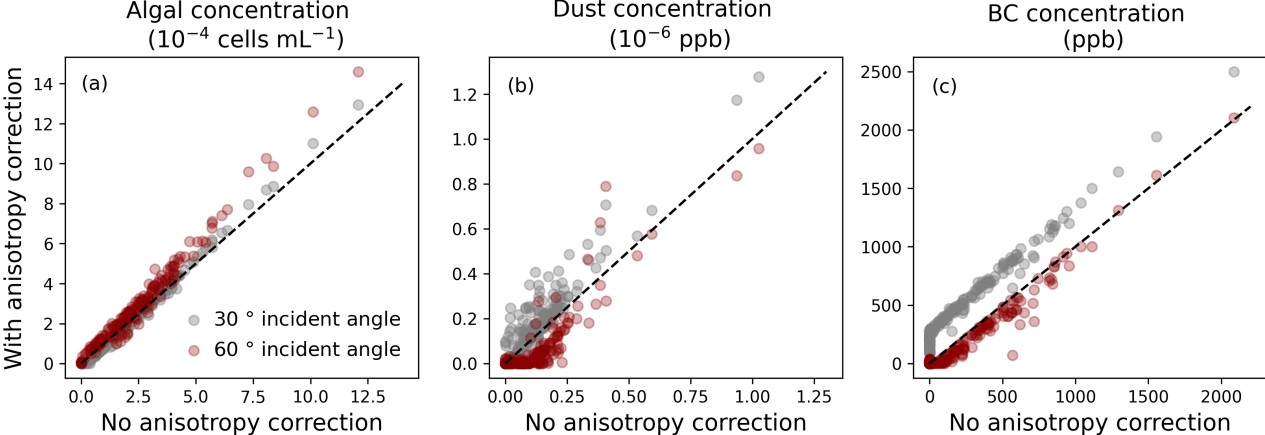

**Figure A2.** Sensitivity tests to evaluate the potential effect of snow anisotropy on the retrievals of (a) algal, (b) dust and (c) black carbon concentrations with the inversion algorithm. The retrievals of the inversions on LAPs concentrations are compared to the retrievals when applying an ARF from Dumont et al. (2010), measured over a flat snow surface with impurities at an incident angle of 30 ° and 60 °.

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
