# Peer review of "Separating the albedo reducing effect of different light absorbing particles on snow using deep learning"

_EGUsphere, 2024_

## Author Comment (AC1)

**Revision of EGUSPHERE-2024-2583 - RC1**

We thank Niklas Bohn for a thorough and encouraging feedback on our manuscript, which improved the clarity and justification of the presented work. We address the different comments below, with our answers highlighted in purple.

**General comments**

**General comment 1** – A short introduction to BioSNICAR/SNICAR would be good. What are its basic concepts? Which input options exist? What's the output quantity? Which radiative transfer approach is used (two-stream vs. multi-stream)? In particular, Section 2.1.1 would better go as an introduction to the utilized radiative transfer model, i.e., BioSNICAR, with changing the section title accordingly. You could still keep the description of input and output data, but expand a bit more on the underlying physics of SNICAR.

**Reply**: We have incorporated this suggestion and updated section 2.1.1 with a new title "Radiative transfer model simulations" and a general description of BioSNICAR:
"The RTM BioSNICAR simulates the bi-hemispherical albedo of snow and ice surfaces by solving the two-stream radiative transfer equations. The model considers the snowpack as homogeneous and plane parallel, and an infinite number of layers with varying snow grain size and shape, density and light absorbing particle concentrations can be prescribed. Several types of incident irradiance can be selected in the model, notably varying with the solar zenith angle. The capabilities and physical equations of the model are similar to the latest SNICAR version, and are detailed in Flanner et al. [2021] and Whicker et al. [2022]. Here, the RTM was parametrised with [...]"

**General comment 2** – You represent snow as a granular medium with spherical grains. However, many controversies exist in the literature about how to model the shape of snow grains. I usually apply the 'collection of spheres' approach from Grenfell and Warren (1999) myself, but a brief discussion about potential impacts of assuming the spherical representation, and possible alternatives would be good.

**Reply**: We agree that a discussion on this assumption was needed, and we added new sentences in the section 2.1.1 to justify our approach and mention alternatives:
"Snow grains were represented by spheres as per the original formulation of the SNICAR model [Flanner et al., 2021, Wiscombe and Warren, 1980]. Recent work showed that light penetration in snow is better represented using irregularly shaped grains [e.g. Robledano et al., 2023], notably yielding more accurate retrievals of snow specific surface area (SSA), but here we chose to use spherical grains mainly because 1) one objective of this study was to incorporate liquid water in snow using the validated framework of Donahue et al. [2022], which is based on spherical grains, and 2) the main focus of the study was the retrieval of light absorbing particles rather than snow physical properties, hence spherical-equivalent snow SSA were deemed appropriate. Future developments of the emulator may consider more realistic physical representations of snow such as a collection of hexagonal plates [Whicker et al., 2022], irregularly shaped grains [Picard and Libois, 2024], or a random mixture of ice and air phases characterized by their mean chords [Malinka, 2023]."

**General comment 3** – There are different sets of dust optical properties available in SNICAR, depending on their source and/or sampling region. Please provide more detail about which exact dataset you used and why.

**Reply**: We have used the optical properties generated by Skiles et al. [2017] with the size range 10-50 $\mu$m because large dust particles are often found in snowpacks [Flanner et al., 2021, Skiles et al., 2017]. This has been clarified in the methods:
"For the light absorbing particles, the optical properties of black carbon (BC) and dust were directly available in the model. For BC, we used the uncoated BC optical properties from Flanner et al. [2012] and for the dust we used the optical properties of the dust mixture from Skiles et al. [2017] in the size bin 10-50 $\mu$m, which are representative of measurements from dust in snowpacks [Flanner et al., 2021, Skiles et al., 2017]."

**General comment 4** – It would be good to have a paragraph in the introduction highlighting the potential of hyperspectral/multispectral sensors for remote sensing of LAP in snow.

**Reply**: We agree that mentioning the relevance of remote sensors was missing from the introduction so we added the sentence copied below, however we decided not to include a full paragraph so as to not deviate from the focus of the study, which does not include any analysis of remotely sensed data.
"[In addition, inverse methods offer a remote-only approach to detecting and quantifying LAPs in hard-to-access areas and/or over spatial scales that are too large to cover on foot]. They are hence particularly relevant in the context of ongoing (e.g. PRISMA, EnMAP) and emerging (e.g., SBG) satellite missions providing remotely sensed reflectance imagery at high spectral resolution."

**Specific comments**

**Specific comment 1, lines 25 - 26** – Can you provide any references to studies that did these forward modeling experiments?

**Reply**: We are unfortunately not aware of forward modeling experiments that quantified uncertainties related to the high number of free parameters in RTMs. However, several modeling studies including for example Flanner et al. [2021] or He [2022], referenced in the sentence above, have demonstrated the large difference in the simulated BBA for e.g. varying snow grain shape, snow algae pigmentation or cell size, highlighting the potential errors arising from fixing these parameters arbitrarily.

**Specific comment 2, line 37** – Bohn et al. (2021) do not actually use Gaussian processes, but simply optimal estimation that acts on the assumption that state parameters and their errors show a Gaussian distribution.

**Reply**: Apologies for the error, we have corrected the sentence: "To our knowledge, only one study has developed an inverse method to discriminate between biotic and abiotic LAPs, using optimal estimation Bohn et al. [2021]."

**Specific comment 3, line 45** – Why snow algae in particular? From my experience, modeling the effects of black carbon particles is clearly more challenging because of their minor absorptivity and weak occurrences.

**Reply**: The sentence line 45 mentioned the need to improve and validate the representation of LAPs in albedo models, and hence we mentioned specifically snow algae because very few studies have focused

on the development and validation of albedo models for snow algal albedo effect, in comparison to black carbon. This sentence notably links to the conclusion of the abstract of the SNICAR paper by Flanner et al. [2021]: "More work is needed particularly in the representation of snow algae, including experimental verification of how different pigment expressions and algal cell concentrations affect snow albedo.". We agree that modeling the effect of BC is also challenging, but in our understanding this has less to do with the validity of the albedo model than the uncertainties on the presence of BC, and its overall low concentrations yielding subtle signatures that are difficult to reliably detect and model.

**Specific comment 4, lines 59 − 61 −** I don't think you need to mention all the used Python libraries here. Maybe only tensorflow and keras as those two are important for understanding the applied deep learning approach.

**Reply**: We have moved the sentence enumerating the python libraries to the acknowledgments to ease readability, but we would like to keep the sentence in the manuscript so as to acknowledge the open source developers who contributed to create the libraries used in our analysis.

**Specific comment 5, line 73 −** To better justify this, you could create non-linear but homogeneous grids by using Sobol sequences (`https://docs.scipy.org/doc/scipy/reference/generated/scipy.stats.qmc.Sobol.html`).

**Reply**: Thank you for the suggestion, we will definitely consider using this tool for future studies!

**Specific comment 6, Table 1 −** Why only going up to 15%? Previous studies have shown that up to 25% could be realistic (Green et al. 2002; Bohn et al. 2021).

**Reply**: The training dataset included simulations with a liquid water content (LWC) only until 15% but the emulator can extrapolate beyond the training dataset and simulate the albedo with higher LWCs that accurately reproduce the original model (see figure below, the dashed lines being the emulator and the solid lines being BioSNICAR).

[Figure]

**Specific comment 7, line 92 −** Can you further substantiate this assumption?

**Reply**: We have added a new reference and modified the sentence to: "The optical properties are representative of algal bloom from the Greenland ice sheet where the samples were collected [Chevrollier

et al., 2023], and are assumed to generalise to all red algal blooms since red snow is caused by only a few species sharing a similar pigmentation [Lutz et al., 2016]."

**Specific comment 8, line 94-95** − Hyperspectral/multispectral comes out of the blue here. See general comments.

**Reply**: We have changed the sentence to "(2) radiometric sensors on ground, airborne or spaceborne platforms rarely detect signals above 2.5 $\mu$m".

**Specific comment 9, line 123** − How fast is this inversion scheme? Could you give a few numbers?

**Reply**: It is difficult to give precise numbers since this highly depends on the machine, the version of tensorflow and the spectral resolution of the data, but we tried to be more precise line 190: "At present, the inversion method is fast enough to be scaled to satellite observations, with a computation time of about 50mn to invert 100km$^2$ of Sentinel-2 imagery at 60m resolution (tensorflow v2.16.0, CPU AMD Ryzen 7 7700X and RAM 64Gb)."

**Specific comment 10, line 134** − What does 'homogeneous enough' mean? Did you have minor influences from other surface types or roughness?

**Reply**: In this context, homogeneous enough meant that the surfaces "seen" by the different fibers in the FieldSpec were similar enough so as to not produce a discontinuity. This issue is described in Painter [2011]: "The optic cable of the ASD (Analytical Spectral Devices, Inc.) Field Spec spectroradiometer has an anisotropic distribution of the wavelength-dependent fibers that creates a sampling scenario in which different areas of the surface are observed with different parts of the spectrum. Without a randomizing filter, this often results in stepwise differences between the ASD FieldSpec VNIR (Si), SWIR1 (InGaAs) and SWIR2 (InGaAs) spectrometers that have wavelength ranges of 350–1000 nm, 1001–1800 nm and 1801–2500 nm, respectively." We have added the reference to Painter [2011] in the sentence to clarify the meaning of the sentence.

**Specific comment 11, line 145** − Calculated radiative forcing is not instantaneous when you use 24h daily averaged shortwave incoming radiation. To get the instantaneous radiative forcing, you would need to multiply the BBA reduction by the incoming radiation at the exact time of the measurement.

**Reply**: Indeed, we had omitted to describe the instantaneous radiative forcing in the section 2.3 and corrected accordingly:
"The daily and instantaneous radiative forcings (W m$^{-2}$) were calculated by multiplying the BBA reduction with respectively the 24h daily averaged and instantaneous shortwave incoming radiation, as measured with a four-component radiometer (CNR4, Kipp and Zonen, The Netherlands) at the local weather station [Pirk et al., 2023]."

**Specific comment 12, line 195** − You need to clarify a bit better if you applied the ARFs to the HCRF spectra before doing the inversions. I guess you did not, but it's not fully clear from the text.

**Reply**: We have clarified this sentence to indicate that the inverted quantities correspond to the direct measurements: "The inversion algorithm successfully reproduced the ground HCRF spectra measured on snowfields in Southern Norway [..]".

**Specific comment 13, line 208-210** − So why not using a multi-stream RTM such as DISORT to account for reflectance anisotropy?

**Reply**: The reason we chose the two-stream model BioSNICAR in this study was because 1) it already has several features of interest built-in and 2) to the best of our understanding the anisotropy of the surface highly depends on its properties and it is unclear how well the BRDF of a collection of spheres as prescribed in DISORT can represent the BRDF of densely packed, wet and contaminated snow, which is why we used empirical coefficients instead to discuss the effect of anisotropy. On the other hand, an important aim of this study was to provide a framework replicable with any RTM, and we agree that using a model that may be more advanced than BioSNICAR/SNICAR would be a significant upgrade.

**Specific comment 14, Figure 2** − I don't see the values of the retrieved grain radius in these figures.

**Reply**: We originally decided to omit the grain size values in the figures to ease readability and reduce the size of the legend since it was not the focus of the study, but we have now changed the legend to add the grain size.

**Specific comment 15, line 239** − You need to clarify in the methods section how you calculated the IRF.

**Reply**: This was addressed in specific comment 11.

**Specific comment 16, line 253** − Again, please clarify at the beginning, which type of dust OPs you're applying in this study. See general comments.

**Reply**: We have clarified this in the methods as per our answer in the general comment 3.

**Specific comment 17, line 256** − The expression light absorbing particles darkening snow surfaces sounds odd, please try to revise.

**Reply**: We have revised as "Light absorbing particles have varying apparent optical properties, challenging the quantification of their albedo reducing effect using forward modeling and making inverse methods relevant for their detection."

**Specific comment 18, Figure A1** − : Again, the grain size values are missing in this plot.

**Reply**: We originally decided to omit the grain size values in the figures to ease readability and reduce the size of the legend since it was not the focus of the study, but we have now changed the legend to add the grain size.

**Technical corrections**

**Line 1**: Several different types of light absorbing particles (LAPs) darken snow surfaces, . . .

**Reply**: Corrected.

**Line 49**: BioSNICAR

**Reply**: Corrected.

**Line 56**: BioSNICAR

**Reply**: Corrected.

**Table 1**: The caption should be located above the table.

**Reply**: Corrected.

**Line 93**: spectral albedo

**Reply**: Corrected.

**Line 134**: an hour prior **to** the measurements ...

**Reply**: Corrected.

**Line 140**: Since the measurements are not equivalent to ...

**Reply**: Corrected.

**Line 148**: the spectral albedo output by the ...

**Reply**: Corrected.

**Line 171**: The emulator is therefore a practical ...

**Reply**: Corrected.

**Figure 1, caption**: ... to the (b) highest and (c) lower mean ...

**Reply**: Corrected.

**Table 2**: Again, I think the caption goes above the table.

**Reply**: Corrected.

**Line 216**: if the spectral diffuse and direct partitioning ...

**Reply**: Corrected.

**Figure 2, caption**: ... is approximately 45x45 cm, centered on ...

**Reply**: Corrected.

**Line 252**: ... could be integrated in the ...

**Reply**: Corrected.

**Line 259**:... emulating a radiative transfer model, and an ...

**Reply**: Corrected.

**Figure A1, caption**: ... is approximately 45x45 cm, centered on ...

**Reply**: Corrected.

**Figure A2, caption**: The retrievals of LAP concentrations are compared to the retrievals when applying ARFs from ...

**Reply**: Corrected.

**References**

N. Bohn, T. H. Painter, D. R. Thompson, N. Carmon, J. Susiluoto, M. J. Turmon, M. C. Helmlinger, R. O. Green, J. M. Cook, and L. Guanter. Optimal estimation of snow and ice surface parameters from imaging spectroscopy measurements. *Remote Sensing of Environment*, 264:112613, 2021.

L.-A. Chevrollier, J. M. Cook, L. Halbach, H. Jakobsen, L. G. Benning, A. M. Anesio, and M. Tranter. Light absorption and albedo reduction by pigmented microalgae on snow and ice. *Journal of Glaciology*, 69(274):333–341, 2023.

C. Donahue, S. M. Skiles, and K. Hammonds. Mapping liquid water content in snow at the millimeter scale: an intercomparison of mixed-phase optical property models using hyperspectral imaging and in situ measurements. *The Cryosphere*, 16(1):43–59, 2022.

M. Flanner, X. Liu, C. Zhou, J. E. Penner, and C. Jiao. Enhanced solar energy absorption by internally-mixed black carbon in snow grains. *Atmospheric Chemistry and Physics*, 12(10):4699–4721, 2012.

M. G. Flanner, J. B. Arnheim, J. M. Cook, C. Dang, C. He, X. Huang, D. Singh, S. M. Skiles, C. A. Whicker, and C. S. Zender. Snicar-adv3: a community tool for modeling spectral snow albedo. *Geoscientific Model Development*, 14(12):7673–7704, 2021.

C. He. Modelling light-absorbing particle–snow–radiation interactions and impacts on snow albedo: fundamentals, recent advances and future directions. *Environmental Chemistry*, 19(5):296–311, 2022.

S. Lutz, A. M. Anesio, R. Raiswell, A. Edwards, R. J. Newton, F. Gill, and L. G. Benning. The biogeography of red snow microbiomes and their role in melting arctic glaciers. *Nature communications*, 7(1):11968, 2016.

A. Malinka. Stereological approach to radiative transfer in porous materials. application to the optics of snow. *Journal of Quantitative Spectroscopy and Radiative Transfer*, 295:108410, 2023.

T. H. Painter. Comment on singh and others,'hyperspectral analysis of snow reflectance to understand the effects of contamination and grain size'. *Journal of Glaciology*, 57(201):183–185, 2011.

G. Picard and Q. Libois. Simulation of snow albedo and solar irradiance profile with the two-stream radiative transfer in snow (tartes) v2. 0 model. *Geoscientific Model Development*, 17(24):8927–8953, 2024.

N. Pirk, K. Aalstad, Y. A. Yilmaz, A. Vatne, A. L. Popp, P. Horvath, A. Bryn, A. V. Vollsnes, S. Westermann, T. K. Berntsen, F. Stordal, and L. M. Tallaksen. Snow–vegetation–atmosphere interactions in alpine tundra. *Biogeosciences*, 20(11):2031–2047, June 2023. ISSN 1726-4189. doi: 10.5194/bg-20-2031-2023. URL `http://dx.doi.org/10.5194/bg-20-2031-2023`.

A. Robledano, G. Picard, M. Dumont, F. Flin, L. Arnaud, and Q. Libois. Unraveling the optical shape of snow. *Nature Communications*, 14(1):3955, 2023.

S. M. Skiles, T. Painter, and G. S. Okin. A method to retrieve the spectral complex refractive index and single scattering optical properties of dust deposited in mountain snow. *Journal of Glaciology*, 63(237):133–147, 2017.

C. A. Whicker, M. G. Flanner, C. Dang, C. S. Zender, J. M. Cook, and A. S. Gardner. Snicar-adv4: a physically based radiative transfer model to represent the spectral albedo of glacier ice. *The Cryosphere*, 16(4):1197–1220, 2022.

W. J. Wiscombe and S. G. Warren. A model for the spectral albedo of snow. i: Pure snow. *Journal of the Atmospheric Sciences*, 37(12):2712–2733, 1980.

---

## Author Comment (AC2)

**Revision of EGUSPHERE-2024-2583 - RC2**

We thank the reviewer for a constructive and encouraging feedback, which helped clarify the method and identify different avenues for future development. We address the different comments below, with our answers highlighted in purple.

**Major comments**

**Major comment 1** − I am a little confused by the necessity of developing/using the machine learning (ML) emulator of RTM. Does the optimising retrieval algorithm only work with the ML emulator? Can the algorithm also work/couple directly with the physics-based RTM e.g., BioS-NICAR? Or is it just for the computational efficiency purpose?

**Reply**: The necessity of developing an ML emulator in the context of this study was indeed a matter of computational efficiency, because applying an optimization algorithm to the original model would have been very slow/resource demanding. Emulating the RTM with a neural network improved the efficiency by 1) accelerating the forward runs prior gradient computation and 2) accelerating the gradient computation by leveraging automatic differentiation via TensorFlow [Jouvet et al., 2021]. The algorithm in its current form therefore requires the model to be written/readable by TensorFlow, and hence cannot be applied directly to BioSNICAR, which is written with more common python libraries such as numpy, that do not enable automatic differentiation.
We have changed section 2.1 to clarify this point:
"The inversion scheme is based on the open source RTM BioSNICAR [Cook et al., 2020], a python translation of the SNICAR model [Flanner et al., 2021]. Directly optimizing BioSNICAR via a gradient-descent algorithm would have been too computationally expensive and hence we built and used a deep learning neural network emulator of the RTM in order to improve the efficiency of the inversion, and notably 1) accelerate the forward runs used in the optimisation algorithm prior the gradient computation and 2) accelerate the gradient computation by leveraging the automatic differentiation framework of TensorFlow. A training dataset of simulations of the original RTM was first generated (section 2.1.1) and used to build the emulator (section 2.1.2), which was then coupled to an optimising algorithm (2.1.3) to invert spectral albedo for surface properties, including the darkening effect of light absorbing particles."

**Major comment 2** − The authors made a few key assumptions when creating the training dataset, such as two snow layers, spherical snow grains, only upper layer for LAPs, and constant snow density, which limit the applicability of the emulator. Among these assumptions, the top 2 cm snow layer containing LAPs and constant density are probably two most important limitations, which could be relaxed to allow them to vary during the emulator training to increase the applicability of the emulator for future studies. Particularly, only the top 2 cm containing LAPs is not realistic.

**Reply**: We agree with the reviewer that a fixed configuration of the RTM in principle limits the applicability of the emulator, but we would argue that there are a number of advantages in proposing a simplified configuration, notably because the number of free parameters as input of the emulator must be balanced with both the available knowledge to constrain these parameters and the simplifications inherent to the physical model. In this study, the choice of parameters results from an extensive manual

testing to understand the sensitivity of the model to each parameter and subsequent identification of an optimal configuration that captures the processes of interest (changes in SSA, LWC, and concentrations of LAPs) whilst reducing the number of free parameters. In the paper, we show that the emulator is able to reproduce the signature of a wide range of surface conditions, and hence we argue that this limited configuration may be helpful for forward modeling experiments to avoid tuning a large number of parameters.

We clarify our reasoning regarding the different parameters fixed in the training dataset below, in particular for the depth and density, and hope that it provides a better justification for the design choices of the study and future avenues for development.

**Depth of light absorbing particles** We agree with the reviewer that this is a simplification and that the depth of light absorbing particles varies in practice, both during the season and between the types of light absorbing particles, hence enabling the emulator to model LAPs in several layers is an important avenue for future developments. Here we decided to fix it to 2cm because 1) it is the depth commonly reported for snow algal studies, 2) the effect of LAPs below 2cm can be modeled with equivalent concentrations in the upper 2cm layer, 3) the size of the training dataset would have grown too large with additional configurations of BioSNICAR and 4) our main goal in this study was to quantify the impact of LAPs by inversion, which is not affected by the depth at which the LAPs are implemented in the model because the inversion algorithm will simply adapt the retrieved concentration to match the apparent properties of the LAPs. As such, the concentration of LAPs in the emulator must be understood as a "2cm-equivalent", which we clarified in the text: "A 2 cm depth was chosen for the upper layer as this depth was used to quantify algal cells in recent field studies [Engstrom et al., 2022, Healy and Khan, 2023], hence the LAP concentrations represent 2cm-equivalents".

**Snow density** In SNICAR/BioSNICAR, snow is represented as a collection of grains and the effective size of the grains essentially drives the effect of the snow physical properties on the spectral albedo [Flanner et al., 2021, Wiscombe and Warren, 1980]. The effect of the density on the spectral albedo is in comparison minimal, which is why we decided to fix the snow density to an average value and work solely with the SSA. We have clarified this in the methods:

"The density was kept constant at 600 kg m$^{-3}$, because it minimally impacts the spectral albedo in comparison to the snow grain size, which is here an effective optical grain size [Gardner and Sharp, 2010, Warren, 1982] that covers realistic ranges of snow specific surface area for melting snow (1-10 m2 kg$^{-1}$; Dumont et al. [2017], Tuzet et al. [2020])."

We agree with the reviewer that in practice snow density is an important physical variable determining snow albedo, and other physical representations of snow may better account for it [e.g. Malinka, 2023], also offering improvement avenues for the emulator.

**Snow layers** We agree with the reviewer that snow columns contain many snow layers with varying densities in reality, and here we chose to restrict the configuration to two layers mainly as a matter of computational resources. If we had increased the number of layers while allowing the SSA to change in each layer, the size of the training dataset would basically have become unreasonably large both in terms of storage and RAM to train the emulator. This is even more true if the SSA + the LAPs were to vary in the layers. In contrast, if we had added more layers but fixed the variables in the layers, then the resulting configuration would have been equivalent to the current configuration. A possible computationally-efficient improvement avenue to have more layers would be to parametrise the SSA of

the snow column as a function of the SSA of the first layer, and we are hoping to explore this possibility in future work.

**Spherical snow grains**   We agree with the reviewer that this assumption limits the applicability of the emulator, as also pointed out by Reviewer 1. We have now added a paragraph to discuss this issue and mention alternatives:

"Snow grains were represented by spheres as per the original formulation of the SNICAR model [Flanner et al., 2021, Wiscombe and Warren, 1980]. Recent work showed that light penetration in snow is better represented using irregularly shaped grains [e.g. Robledano et al., 2023], notably yielding more accurate retrievals of snow specific surface area (SSA), but here we chose to use spherical grains mainly because 1) one objective of this study was to incorporate liquid water in snow using the validated framework of Donahue et al. [2022], which is based on spherical grains, and 2) the main focus of the study was the retrieval of light absorbing particles rather than snow physical properties, hence spherical-equivalent snow SSA were deemed appropriate. Future developments of the emulator may consider more realistic physical representations of snow such as a collection of hexagonal plates [Whicker et al., 2022], irregularly shaped grains [Picard and Libois, 2024], or a random mixture of ice and air phases characterized by their mean chords [Malinka, 2023]."

**Major comment 3** –   Does the inversion algorithm search for local optima or global optima? Would there be equifinality issue? Also, how sensitive is the algorithm retrieval result to the initial guess and how to effectively select the initial guess?

**Reply**:   The inversion algorithm is a simple stochastic gradient descent algorithm, and hence it does not particularly search for a global optima. However, the tests conducted in section 3.1.2 show that the algorithm is able to retrieve the optimal solution (global minima) from a simulated spectrum (i.e, the error in the retrieved parameters is negligible) and this was verified for 20 independent pseudo-random intial guesses. The application of the algorithm to ground measurements in section 3.2 then shows that the algorithm retrieves a consistent and quasi-unique solution for a measured spectrum (i.e. negligible variation between the 20 retrievals; line 205-208). Hence, the equifinality and sensitivity to the intial guess do not seem to be an issue in our study, as the algorithm reaches the same optimal solution regardless of the initial guess.

**Minor comments**

**Minor comment 1, line 95**: What is the reason for the biosnicar discontinuity around 2.5um?

**Reply**:   We do not know exactly what causes this discontinuity, but known issues exist with for example with the Eddington solver for specific combinations of parameters [Toon et al., 1989], and a similar issue may cause these numerical errors.

**Minor comment 2, line 223-250**: How did the authors compute the radiative forcing from albedo reduction? What downward solar radiation data did the authors use?

**Reply**:   The downward solar radiation used was directly measured at the local weather station, as described in the methods section 2.3, which has been changed for better clarity:
"The daily and instantaneous radiative forcings (W m$^{-2}$) were calculated by multiplying the BBA reduction with respectively the 24h daily averaged and instantaneous shortwave incoming radiation, as

measured with a four-component radiometer (CNR4, Kipp and Zonen, The Netherlands) at the local weather station [Pirk et al., 2023]."

**Minor comment 3**: I would suggest adding a section to discuss uncertainties involved in the ML-based emulator and the inversion algorithm.

**Reply**: We have dedicated a specific result section to the description of the errors of the neural network in the training and validation (3.1.1) as well as the retrievals of the inversion algorithm on both simulated spectra (3.1.2) and measured spectra (3.1.3), hence we are not sure how to add further discussion on the uncertainties of the emulator and inversion algorithm in the present manuscript.

**References**

J. M. Cook, A. J. Tedstone, C. Williamson, J. McCutcheon, A. J. Hodson, A. Dayal, M. Skiles, S. Hofer, R. Bryant, O. McAree, et al. Glacier algae accelerate melt rates on the south-western greenland ice sheet. *The Cryosphere*, 14(1):309–330, 2020.

C. Donahue, S. M. Skiles, and K. Hammonds. Mapping liquid water content in snow at the millimeter scale: an intercomparison of mixed-phase optical property models using hyperspectral imaging and in situ measurements. *The Cryosphere*, 16(1):43–59, 2022.

M. Dumont, L. Arnaud, G. Picard, Q. Libois, Y. Lejeune, P. Nabat, D. Voisin, and S. Morin. In situ continuous visible and near-infrared spectroscopy of an alpine snowpack. *The Cryosphere*, 11(3):1091–1110, 2017.

C. B. Engstrom, S. N. Williamson, J. A. Gamon, and L. M. Quarmby. Seasonal development and radiative forcing of red snow algal blooms on two glaciers in british columbia, canada, summer 2020. *Remote Sensing of Environment*, 280:113164, 2022.

M. G. Flanner, J. B. Arnheim, J. M. Cook, C. Dang, C. He, X. Huang, D. Singh, S. M. Skiles, C. A. Whicker, and C. S. Zender. Snicar-adv3: a community tool for modeling spectral snow albedo. *Geoscientific Model Development*, 14(12):7673–7704, 2021.

A. S. Gardner and M. J. Sharp. A review of snow and ice albedo and the development of a new physically based broadband albedo parameterization. *Journal of Geophysical Research: Earth Surface*, 115(F1), 2010.

S. M. Healy and A. L. Khan. Albedo change from snow algae blooms can contribute substantially to snow melt in the north cascades, usa. *Communications Earth & Environment*, 4(1):142, 2023.

G. Jouvet, G. Cordonnier, B. Kim, M. Lüthi, A. Vieli, and A. Aschwanden. Deep learning speeds up ice flow modelling by several orders of magnitude. *Journal of Glaciology*, 68(270):651–664, Dec. 2021. ISSN 1727-5652. doi: 10.1017/jog.2021.120. URL `http://dx.doi.org/10.1017/jog.2021.120`.

A. Malinka. Stereological approach to radiative transfer in porous materials. application to the optics of snow. *Journal of Quantitative Spectroscopy and Radiative Transfer*, 295:108410, 2023.

G. Picard and Q. Libois. Simulation of snow albedo and solar irradiance profile with the two-stream radiative transfer in snow (tartes) v2. 0 model. *Geoscientific Model Development*, 17(24): 8927–8953, 2024.

N. Pirk, K. Aalstad, Y. A. Yilmaz, A. Vatne, A. L. Popp, P. Horvath, A. Bryn, A. V. Vollsnes, S. Westermann, T. K. Berntsen, F. Stordal, and L. M. Tallaksen. Snow–vegetation–atmosphere interactions in alpine tundra. *Biogeosciences*, 20(11):2031–2047, June 2023. ISSN 1726-4189. doi: 10.5194/bg-20-2031-2023. URL http://dx.doi.org/10.5194/bg-20-2031-2023.

A. Robledano, G. Picard, M. Dumont, F. Flin, L. Arnaud, and Q. Libois. Unraveling the optical shape of snow. *Nature Communications*, 14(1):3955, 2023.

O. B. Toon, C. McKay, T. Ackerman, and K. Santhanam. Rapid calculation of radiative heating rates and photodissociation rates in inhomogeneous multiple scattering atmospheres. *Journal of Geophysical Research: Atmospheres*, 94(D13):16287–16301, 1989.

F. Tuzet, M. Dumont, G. Picard, M. Lamare, D. Voisin, P. Nabat, M. Lafaysse, F. Larue, J. Revuelto, and L. Arnaud. Quantification of the radiative impact of light-absorbing particles during two contrasted snow seasons at col du lautaret (2058 m asl, french alps). *The Cryosphere*, 14 (12):4553–4579, 2020.

S. G. Warren. Optical properties of snow. *Reviews of Geophysics*, 20(1):67–89, 1982.

C. A. Whicker, M. G. Flanner, C. Dang, C. S. Zender, J. M. Cook, and A. S. Gardner. Snicar-adv4: a physically based radiative transfer model to represent the spectral albedo of glacier ice. *The Cryosphere*, 16(4):1197–1220, 2022.

W. J. Wiscombe and S. G. Warren. A model for the spectral albedo of snow. i: Pure snow. *Journal of the Atmospheric Sciences*, 37(12):2712–2733, 1980.

---

## Author Comment (AC3)

**Revision of EGUSPHERE-2024-2583 - RC3**

We thank the reviewer for a positive and encouraging feedback, which helped us clarify the opportunities and challenges associated with scaling-up our method. We address the different comments below, with our answers highlighted in purple.

**Major comments**

**Major comment 1** – A short description in the methods of biosnicar/SNICAR would be useful. This should include the optical schemes used and any error associate with the model.

**Reply**: To address this comment as well as the general comment 1 from reviewer 1, we have changed section 2.1.1 to better describe BioSNICAR:
"The RTM BioSNICAR simulates the bi-hemispherical albedo of snow and ice surfaces by solving the two-stream radiative transfer equations. The model considers the snowpack as homogeneous and plane parallel, and an infinite number of layers with varying snow grain size and shape, density and light absorbing particle concentrations can be prescribed. Several types of incident irradiance can be selected in the model, notably varying with the solar zenith angle. The capabilities and physical equations of the model are similar to the latest SNICAR version, and are detailed in Flanner et al. [2021] and Whicker et al. [2022]. Here, the RTM was parametrised with [. . . ]"

**Major comment 2** – This work is very exciting for potential use in larger scale hyperspectral measurements. The conclusions could use some discussion of any potential issues or challenges of scaling the model up to regional or gloabal scale, especially with the mention of satellite use. For example, dust optical properties can vary by region. Would this impact use of the model?

**Reply**: Thank you for the positive feedback! We have included the following paragraph at the end of the conclusion to discuss the challenges that we could identify, as well as our efforts to promote the usability of the method:
"To facilitate future usage and development of the method, the full code running the model and the inversion was made available into an open-source python package. Ongoing developments currently focus on making the inversion algorithm resolution-agnostic and hence adaptable to several remote sensing products, as well as adding the possibility to prescribe sensor-specific spectral responses. The application of the method to new areas using remotely sensed imagery will present additional challenges to consider, such as (i) the variability in mineral dust optical properties that may require new mineral mixtures in the model, (ii) the presence of shallow snowpacks of which signature could be confounded with that of black carbon (Warren et al., 2019), or (ii) the variability in spectral resolution between sensors, where lower resolution imagery may require stronger constrains on the inverse problem."

**Minor comments**

**Minor comment 1, line 27**: "..allow to study the impacts of LAPs..." is a bit difficult to read, consider rewording.

**Reply**: We have rephrased as "By contrast, inverse modelling approaches consider the impact of LAPs in snow directly from their measured apparent optical properties instead of prescribing all the above parameters, circumventing some of the uncertainty associated with forward modelling experiments."

**Minor comment 2**: In the RTM model setup, two snow layers are used, with the lower layer being a semi-infinite layer. Dust and black carbon tend to be deposited in layers throughout the season, often together. The assumption of the near semi-infinite lower layer can introduce some error during the melt season as these buried LAP layers get close to the surface.

**Reply**: We agree with the reviewer (and reviewer 2) that this configuration is a simplification of the complex distribution of LAPs in the snowpack. In the context of this study however, we aimed at quantifying the impact of LAPs, which is not affected by the depth at which the LAPs are considered because the model will simply adapt the retrieved concentration to match the apparent properties of the LAPs to yield the impact on the BBA. For forward runs, the concentration of LAPs must however be understood as a "2cm-equivalent", which we clarified in the text: "A 2 cm depth was chosen for the upper layer as this depth was used to quantify algal cells in recent field studies [Engstrom et al., 2022, Healy and Khan, 2023], hence the LAP concentrations represent 2cm-equivalents".

**Minor comment 3**: The title of Section 2.3 should be updated to say daily radiative forcing instead of instantaneous based on the methods described. Instantaneous radiative forcing would multiply the BBA reduction by the incoming solar radiation at the time of the measurement.

**Reply**: We had omitted the description of the incoming solar radiation at the time of the measurement in the methods, hence we have corrected the title to "radiative forcing" and added a description of the instantaneous radiative forcing in addition to the daily radiative forcing:
"The daily and instantaneous radiative forcings (W m$^{-2}$) were calculated by multiplying the BBA reduction with respectively the 24h daily averaged and instantaneous shortwave incoming radiation, as measured with a four-component radiometer (CNR4, Kipp and Zonen, The Netherlands) at the local weather station [Pirk et al., 2023]."

**Minor comment 4**: The methods in Section 2.3 could use some more information. When calculating reduction in BBA from a LAP, is a spectrum with the same grain size and other LAP concentrations being used.

**Reply**: We have clarified this point in the methods:
"The BBA reduction associated with a given LAP was calculated by differencing the BBA of the retrieved solution with the BBA calculated with the exact same conditions (grain size, SZA, LWC...) except the concentration of the given LAP, which was set to 0."

**Minor comment 5**: The results mention both daily average and instantaneous radiative forcing. If both are being used, the calculation of instantaneous radiative forcing should be covered in Section 2.3.

**Reply**: We have added the description of the calculation of instantaneous radiative forcing in Section 2.3 as explained in the minor comment 3.

**References**

C. B. Engstrom, S. N. Williamson, J. A. Gamon, and L. M. Quarmby. Seasonal development and radiative forcing of red snow algal blooms on two glaciers in british columbia, canada, summer 2020. *Remote Sensing of Environment*, 280:113164, 2022.

M. G. Flanner, J. B. Arnheim, J. M. Cook, C. Dang, C. He, X. Huang, D. Singh, S. M. Skiles, C. A. Whicker, and C. S. Zender. Snicar-adv3: a community tool for modeling spectral snow albedo. *Geoscientific Model Development*, 14(12):7673–7704, 2021.

S. M. Healy and A. L. Khan. Albedo change from snow algae blooms can contribute substantially to snow melt in the north cascades, usa. *Communications Earth & Environment*, 4(1):142, 2023.

N. Pirk, K. Aalstad, Y. A. Yilmaz, A. Vatne, A. L. Popp, P. Horvath, A. Bryn, A. V. Vollsnes, S. Westermann, T. K. Berntsen, F. Stordal, and L. M. Tallaksen. Snow–vegetation–atmosphere interactions in alpine tundra. *Biogeosciences*, 20(11):2031–2047, June 2023. ISSN 1726-4189. doi: 10.5194/bg-20-2031-2023. URL `http://dx.doi.org/10.5194/bg-20-2031-2023`.

C. A. Whicker, M. G. Flanner, C. Dang, C. S. Zender, J. M. Cook, and A. S. Gardner. Snicar-adv4: a physically based radiative transfer model to represent the spectral albedo of glacier ice. *The Cryosphere*, 16(4):1197–1220, 2022.

---

## Referee Report (RR1)

Chevrollier et al.: *"Separating the albedo reducing effect of different light absorbing particles on snow using deep learning"*, Review by Niklas Bohn.

I thank the authors for thoroughly addressing all my comments. The manuscript reads more clearly now and shows an improved overall quality. Therefore, I recommend publication in The Cryosphere as is.